# Bionic Plasmonic Nanoarrays Excited by Radially Polarized Vector Beam for Metal-Enhanced Fluorescence

**DOI:** 10.3390/nano13071237

**Published:** 2023-03-31

**Authors:** Min Liu, Lan Yu, Yanru Li, Ying Ma, Sha An, Juanjuan Zheng, Lixin Liu, Ke Lin, Peng Gao

**Affiliations:** 1School of Physics, Xidian University, Xi’an 710071, China; 2Guangzhou Institute of Technology, Xidian University, Guangzhou 510555, China; 3School of Optoelectronic Engineering, Xidian University, Xi’an 710071, China

**Keywords:** metal-enhanced fluorescence, bionic nanoimprint, plasmonic film, radially polarized vector beam

## Abstract

Metal-enhanced fluorescence (MEF) is an important fluorescence technology due to its ability to significantly improve the fluorescence intensity. Here, we present a new MEF configuration of the bionic nanorod array illuminated by radially polarized vector beam (RVB). The bionic nanorod array is fabricated via a nanoimprinting method by using the wings of the Chinese cicada “meimuna mongolica” as bio-templates, and later coating gold film by ion sputtering deposition method. The MEF performance of the prepared substrate is tested by a home-made optical system. The experiment results show that, in the case of RVB excitation, the intensity of fluorescence is more than 10 times stronger with the nano-imprinted substrate than that with glass. Using the bionic nanoarray as a substrate, the intensity of fluorescence is ~2 times stronger via RVB than that by the linearly polarized beam. In addition, the prepared substrate is verified to have good uniformity.

## 1. Introduction

Fluorescence technology, e.g., fluorescence spectroscopy [1,2], super-resolution fluorescence imaging [3], fluorescence correlation spectroscopy [4], has been widely exploited in fields of biology [5], materials [6], diagnostics [7], chemistry [8], etc. The accessible fluorescence intensity and detection volume largely dominate its detection limits. Metal-enhanced fluorescence (MEF) [9,10,11] has increasingly attracted attention due to its ability to significantly improve the intensity of fluorescence and decrease the detection volume. MEF is primarily based on the localized surface plasmon resonance (LSPR) effect [12] of metallic nanostructures to generate the enhanced electromagnetic (EM) field and strengthen the fluorescence of fluorophores in the vicinity of metallic nanostructures. The enhancement factor of fluorescence intensity typically varies with chemical-component changing of plasmonic substrates [13], the different shapes of metallic nanostructures [14], the illumination methods [15], and the distance between fluorophores and metallic nanostructures [16].

The plasmonic substrate is the key to MEF technology. Noble metals, i.e., gold (Au), silver (Ag), copper (Cu) are the most commonly used plasmonic materials. This is because they provide an abundance of free electrons to collectively oscillate and couple with incident light. They possess relatively-low losses at visible wavelength compared to metals such as iron, nickel, etc. Noble-metal nanostructures with different shapes including nanospheres [17,18], nonorods [19,20], nanoprisms [21], nanoarrays [22,23,24], etc., have been verified to have considerable enhancement as MEF substrates. Nanoarrays are one of the most commonly used nanostructures because of excellent activity [25], high repeatability [26] and good compatibility [27]. The excellent activity mainly originates from the strong electric field enhancement induced by the gap mode [28,29,30] in nanogaps between adjacent nanostructures. The high repeatability benefits from the rapid progress of nano-fabricating technology and equipment. The good compatibility lies in its easy-to-transfer and avoiding polluting the target analyte compared to the dispersive nanoparticles. Different geometries and structures of nanoarrays [22,23,24] have been proposed to optimize the MEF performance. Nano-fabricating methods including electron beam lithography [31], inductively coupled plasma (ICP) [32], nanoimprinting [33], etc., have been exploited to successfully prepare nanoarrays with different shapes. The nanoimprinting method has attracted wide attention for its easy cooperation, good repeatability, and low cost.

Besides the plasmonic substrate, the illumination methods largely influence the MEF performance [34,35,36]. The radially polarized vector beam (RVB) [37,38,39] is an unconventional illumination light field. It has a radial distribution of polarization direction on the beam cross-section and sub-diffracted focusing spot under tight focusing condition [40]. RVB was proposed to have successfully improved the metallic nanostructure-based Raman scattering intensity [41,42,43]. More importantly, RVB has been applied in two-photon fluorescence microscopy [35,36] for high resolution. Gold-film-coated Kretschmann and Raether coupling configuration [35], and gold-coated tips [36] have been investigated for fluorescence imaging based on RVB-illuminated MEF. Thus, in the field of fluorescence spectroscopy, it is promising to adopt RVB as illumination source to improve the fluorescence intensity. Considering the advantages of nanoarrays (i.e., excellent activity, high repeatability and good compatibility), it is necessary to explore a new configuration by combining nanoarrays with RVB illumination. Seek for an avenue to enhance the MEF intensity through introducing RVB illumination, so as to develop a new platform for metal-enhanced spectroscopy or microscopy. 

In this paper, a bionic plasmonic film is fabricated by the nano-imprinting method and physical deposition method. The nano-imprinting process is carried out by using the wings of the Chinese cicada “meimuna mongolica” as bio-templates. The physical deposition process is coating gold film by ion sputtering deposition process. The size of the plasmonic film is as large as ~1 × 1 cm^2^. The nanostructures on the film are nanorod arrays. The diameter of a single nanorod is ~150 nm. The nanogaps between adjacent nanorods vary from a few nanometers to tens of nanometers. The plasmonic nanoarray is further used as an MEF substrate. The MEF performance is tested by a home-made optical system. The illumination light is RVB generated via a zero-order vortex waveplate (VHWP). Using RVB as an illumination source, the fluorescence intensity of R6G on the bionic plasmonic film is improved more than 10 times compared to that on the glass substrate. Using the nano-imprinted film as substrate, the fluorescence intensity is ~2 times stronger via RVB than that by linearly polarized beam. In addition, the prepared substrate is verified to have good uniformity, revealing that different areas of the substrates possess similar MEF activity. 

## 2. Materials and Methods

### 2.1. Fabrication of MEF Substrate via Bionic Nano-Imprinting

A MEF substrate was fabricated by the bionic nano-imprinting method and physical deposition method. Figure 1a–d show the fabrication process of bionic nanoarrays via the bionic nano-imprinting method. The wings of the Chinese cicada “meimuna mongolica” were used as bio-templates for nano-imprinting. The wings were cut into quadrate segments with a size of ~1 × 1 cm^2^ for later use. The commercial ultraviolet (UV) curing adhesive (Norland optical adhesive, P/N 6801, Cranbury, NJ, USA) was selected, because it possesses chemical and physical robustness after UV illumination. The nano-imprinting process based on the bio-templates includes two imprinting procedures. The first procedure is forming inverse nanostructures based on wings of meimuna mongolica, as shown in Figure 1a,b. The second procedure is imprinting nanostructures based on the in-verse nanostructures, as shown in Figure 1c,d. 

First, the UV-curing adhesive was uniformly coated on the surface of glass by a spin coater with its rotational speed being 500 r/min for 3 min and 3000 r/min for 5 min. Then, the wing segment was slightly put on the UV-curing adhesive on the glass (G_1_) surface, as exhibited in Figure 1a. The UV light (central wavelength, 365 nm) was vertically incident and propagated through the glass onto the UV-curing adhesive, the illumination time of which lasted for 5 s to solidify the liquid. Here, the UV was generated by a commercial electric-torch. The output power and the central wavelength of the electric-torch is 3 W, 365 nm, respectively. Whether the UV light is on or not is controlled by a mechanical switch. Then, separate the wing segment from the cured UV glue, which leads to the inverse nanostructures left on the surface of the UV glue, which is shown in Figure 1b. Next, the UV-curing adhesive was coated uniformly on the surface of another glass (G_2_) by the process, which was the same as the first step, as demonstrated in Figure 1c. The glass G_1_ with inverse nanostructures on its UV glue was slightly put on the UV-curing adhesive on the surface of G_2_. To quantify the slight imprinting operation by weight, the weight should not exceed 100 g. After being illuminated by the UV light for 5 s, the UV glue on G_2_ was cured. Through separating G_1_ from G_2_, the nanostructures that have similar shapes to the wing surface formed on the surface of the UV glue, as exhibited in Figure 1d. Figure 1e shows the fabrication process of MEF substrate via depositing noble metal onto the prepared bionic nanoarrays by physical deposition method. Gold (Au) was selected as the coating material for its excellent plasmonic property and resistance to oxidation. The ion sputtering deposition method [44] was applied to coating Au film on the bionic nanoarrays. The deposition time was 60 s.

### 2.2. Optical System of MEF

The enhancement performance of the nanoimprinted plasmonic films was tested by a home-made optical system. The spectrum obtained by a glass and the Au coated nanoimprinted film were tested, respectively. The cases of linearly polarized beam (LPB) and radially polarized vector beam (RVB) illumination were compared. The sketch map of the SEF examination system based on the nanoimprinted plasmonic films is shown in Figure 2a. A solid-state laser (CW, 25 mW) (MSL-FN-532, Changchun New Industries Optoelectronics Technology Co., Ltd., Changchun, China) with wavelength of 532 nm was used in the home-made system. A laser-line filter (F_1_) was put after the laser to eliminate the side modes. The laser beam propagated through the polarization plate (P), F_1_, and the zero-order vortex waveplate (VWHP) (Thorlabs, WPV10L-532, Newton, MA, USA), where the LPB was transformed into RVB via VWHP. Then, the generated RVB was reflected by the dichroic mirror (DM, Semrock, Di03-R532-t1-25X36, New York, NY, USA), and propagated through a polarization plate (F_x_), and then recorded by a black-and-white charge coupled device (CCD) to test the polarization characteristics of the generated RVB. Figure 2b shows the recording results of CCD, including energy distribution on the beam cross-section of RVB, and the corresponding polarization testing results of the generated RVB. 

Next, the polarization plate F_x_ and the CCD were removed. The generated RVB was reflected by DM onto the surface of the nano-imprinted plasmonic film carried on a three-dimensional translation stage. The MEF signal of target analytes on the MEF substrate was collected via a micro-objective (MO_1_, Zeiss, 100×, NA = 0.75, Jena, Germany) with long working distance and then propagated through DM. Here, the light beam with wavelength larger than 532 nm propagated through DM and that with smaller wavelength were reflected. Then, the MEF signal was reflected by a mirror (M) and then passed through a long-wavelength edge-filter to eliminate the excitation light beam. A micro-objective (MO_2_, 20×, NA = 0.4, Daheng Optics, Beijing, China) was used to couple the MEF signal into a multimode microfiber that links with a spectrometer (Ocean Insight, Maya 2000 Pro, Orlando, FL, USA). The R6G dispersed in alcohol was selected as target analytes. During the experiment, the R6G dispersion was dropped onto the surface of the MEF substrate by a micropipette.

## 3. Results

### 3.1. Nano-Arrays Fabricated by Nanoimprinting

The nanoimprinted plasmonic films based on bio-templates for MEF were prepared by the bionic nano-imprinting method and physical deposition method, as shown in Figure 1a–e. The cicada wings were selected as the bio-templates in this experiment. It is widely known that cicada wings show colorful pattern under the sunlight illumination. It is because there exists nanoarrays on the surface of the wings [45]. Cicada wings have been used for fabrication of SERS substrate [46,47]. It is promising to use the wings as templates to replicate and realize the low-cost nano-fabrication by nanoimprint process. Through nano-imprinting shown in Figure 1, the inverse nanostructures and the copied nanostructures can be obtained in sequence. The inverse nanostructures were obtained after the first imprinting. Figure 3b shows the morphology of the inverse nanostructures. It was characterized by scanning electron microscope (SEM) from the oblique view. The ‘oblique view’ means that the targets are observed from the oblique angle. Here, the inverse nanostructures were coated by a thin gold film before being visualized by SEM, for that SEM requires the target samples possessing good electrical conductivity. It can be seen from Figure 3b that the inverse nanostructures are nanopore arrays. Nanogaps between adjacent nanopores have been formed. The nanostructures on the surface of wings were copied by the second nanoimprinting using the inverse nanostructures as templates. Figure 3c shows the morphology of the copied nanostructures characterized by SEM from the top-down view. The ‘top-down view’ means that the observer is directly above the targets. It can be seen from Figure 3b that the copied nanostructures are nanorod arrays. The diameter of a single nanorod is about 150 nm. The nanogaps between adjacent nanorods vary from a few nanometers to tens of nanometers. It reveals that the plasmonic nanoarrays for MEF are successfully achieved by combining the bionic nano-imprinting and ion sputtering deposition method.

### 3.2. MEF Detection Excited by RVB

The MEF spectra were detected by the optical system shown in Figure 2a. In this experiment, the wavelength of the excitation light is 532 nm. The target analytes are R6G (Aladdin, R105624, Shanghai, China) molecules dispersed in alcohol. The dispersion was dropped onto the surface of the prepared substrate and set aside for about 5 min for natural drying. Then, the substrate was put onto the sample stage for MEF examination. The testing results are exhibited in Figure 4a–d. Figure 4a shows the fluorescence spectrum of R6G (dispersed in alcohol) on the surface of the nanoimprinted substrate excited via RVB (red) and LPB (black), respectively, and that on the surface of common glass (black) excited via RVB. The integration time is 1 ms. The R6G dispersion has a concentration of 10^−4^ M. Figure 4b exhibits the enlarged spectrum of R6G (dispersed in alcohol) on the surface of common glass excited via RVB. It can be seen from Figure 4a that, in the case of RVB excitation, the intensity of fluorescence is more than 10 times stronger with the nano-imprinted substrate than that without substrate. Using the nano-imprinted film as substrate, the intensity of fluorescence is ~2 times stronger via RVB than that by LPB. The improvement is possibly resulted from the stronger component along the axial direction of the nanorods for RVB and the radial polarization distribution compared to LPB. Moreover, considering the size of the prepared substrate is ~1 × 1 cm^2^, it is important to guarantee different areas of the substrates possessing similar MEF activity. Thus, keeping the parameters (e.g., shape, diameter, adjacent gap size, thickness of the gold film) of nanorod units being constant is necessary. Here, ‘uniformity’ is used to describe the characteristic. The MEF activity of eight different positions of the substrate were tested to explore its uniformity. Figure 4c shows the fluorescence spectrum of R6G dispersion enhanced by eight positions of the substrate. Figure 4d exhibits the corresponding intensity at the wavelength of 556 nm. The variations on parameters of the nanorod units are indicated by the intensity fluctuations. As shown in Figure 4c,d, the weak perturbation of the fluorescence intensity reveals good uniformity of the substrate. However, there still exists uncertainty to use eight positions to reflect the uniformity of the whole substrate with size of 1 × 1 cm^2^. The MEF detection results demonstrate that the prepared MEF substrate has high MEF activity. The nanoimprinted film has the ability of enhancing the intrinsic fluorescence of fluorescent materials. The RVB in free space has the ability of improving the MEF enhancement factor, in the case of nanorod arrays with gold coating being substrates.

## 4. Conclusions

In summary, we present a new MEF configuration of the bionic nanorod array illuminated by RVB. The bionic nanorod array is fabricated via bionic nanoimprinting method by using the cicada wings as bio-templates. The MEF substrate is prepared by coating gold film by ion sputtering deposition method onto the bionic nanorod array. The diameter of a single nanorod is ~150 nm. The nanogaps between adjacent nanorods varies from a few nanometers to tens of nanometers. The size of the bionic film is as large as 1 × 1 cm^2^. The MEF performance of the prepared substrate is tested by a home-made optical system, where the illumination light is RVB generated via a VHWP. The experiment results show that in the case of RVB excitation, the intensity of fluorescence is more than 10 times stronger with the nano-imprinted substrate than that with glass as substrate. Using the bionic nanoarray as a substrate, the intensity of fluorescence is ~2 times stronger via RVB than that by LPB. Moreover, the MEF activity of eight different positions of the substrate were tested to explore its uniformity. Although the weak perturbation of the fluorescence intensity reveals good uniformity of the substrate, there still exists uncertainty to use eight positions to reflect the uniformity of the whole substrate with size of 1 × 1 cm^2^. Summing up the above, the RVB-excited nanoarray has the ability of improving the MEF intensity. The prepared MEF substrate has advantages of large area, high MEF activity, good uniformity and compatibility. The fabrication method is low-cost and easy-to-operate. The configuration of the bionic nanorod array illuminated by RVB may be further exploited in fields of fluorescence technology such as super-resolution fluorescence imaging, fluorescence correlation spectroscopy, etc.

## Figures and Tables

**Figure 1 nanomaterials-13-01237-f001:**
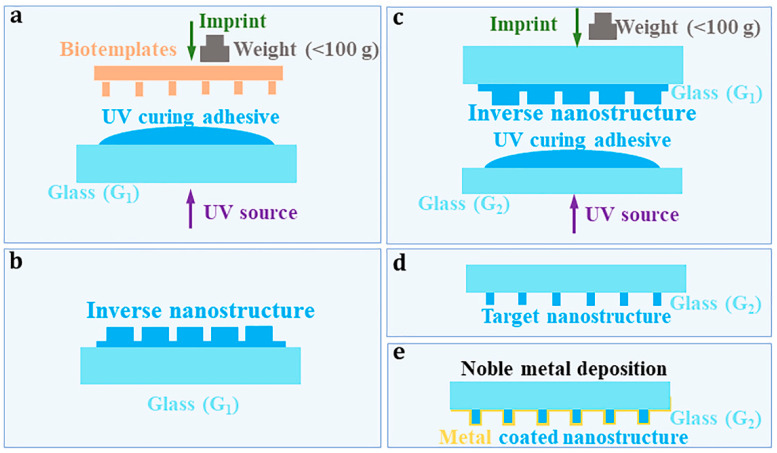
(**a**–**d**) Fabrication process of bionic nanoarrays via the bionic nano-imprinting method by using the cicada wings as bio-templates. (**e**) Fabrication process of MEF substrate via depositing noble metal onto the prepared bionic nanoarrays by physical deposition method.

**Figure 2 nanomaterials-13-01237-f002:**
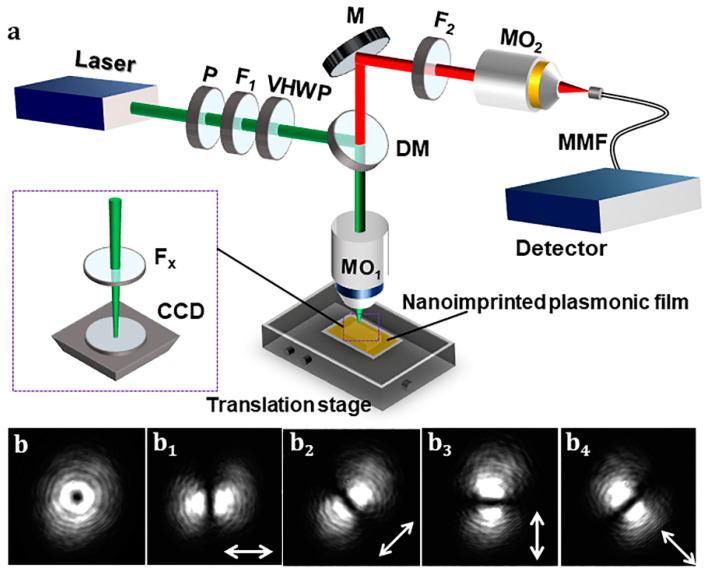
(**a**) Sketch map of the MEF examination system based on the nanoimprinted plasmonic films. (**b**) Energy distribution on the beam cross-section of RVB generated by VHWP, and the corresponding polarization testing results (**b_1_**–**b_4_**), which are obtained by rotating the polarization plate F_x_. The arrows indicate the transmittable polarization direction of F_x_.

**Figure 3 nanomaterials-13-01237-f003:**
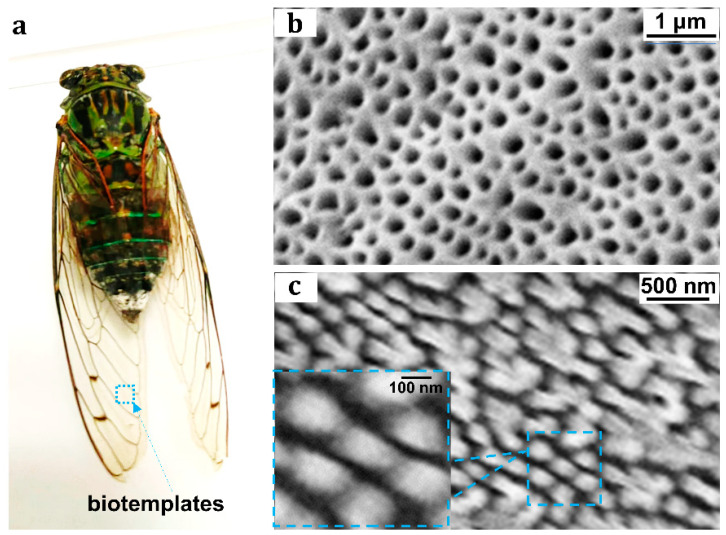
(**a**) Image of Chinese cicada “meimuna mongolica” as bio-template for bionic nano-imprinting. (**b**) The morphology of the inverse nanostructures (after the first imprinting) characterized by scanning electron microscope (SEM). (**c**) The morphology of the copied nanostructures (after the second imprinting) characterized by SEM.

**Figure 4 nanomaterials-13-01237-f004:**
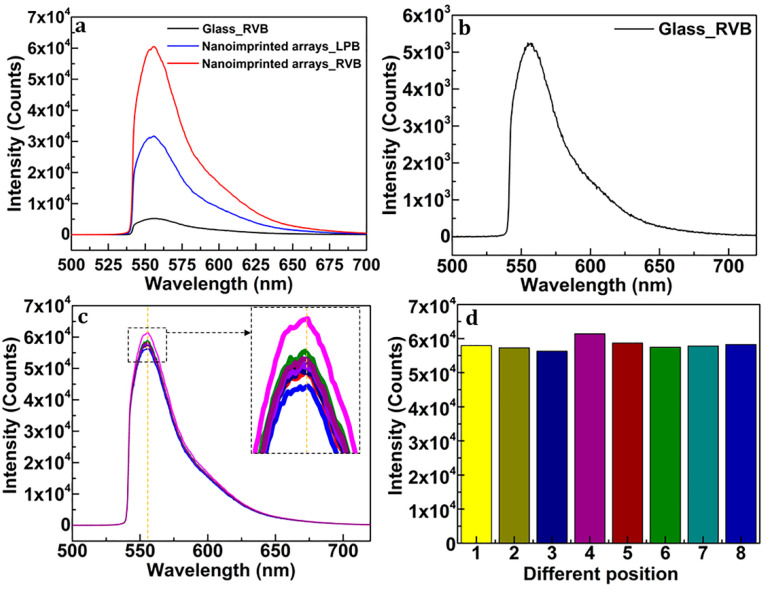
(**a**) The fluorescence spectrum of R6G (dispersed in alcohol) on the sur-face of the nanoimprinted substrate excited via RVB (red) and LPB (blue), respectively, and that on the surface of common glass (black) excited via RVB. The integration time is 1 ms. The R6G dispersion has concentration of 10^−4^ M. (**b**) The enlarged line of the spectrum (black) shown in (**a**). (**c**) The fluorescence spectrum of R6G dispersion enhanced by eight positions of the nanoimprinted substrate. The right subfigure is the enlargement of the area marked by the black dotted box on the left. Different colored solid-lines represent fluorescence spectrum detected at different positions. The orange dotted line is corresponding to the wavelength of 556 nm. (**d**) The corresponding intensity at the wavelength of 556 nm of (**c**).

## Data Availability

Not applicable.

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
