# Peer review of "Bionic Plasmonic Nanoarrays Excited by Radially Polarized Vector Beam for Metal-Enhanced Fluorescence"

_nanomaterials, 2023, doi:10.3390/nano13071237_

Round 1

Reviewer 1 Report

The paper titled "Metal-Enhanced Fluorescence Using Bionic Nanorod Arrays Fabricated by Nanoimprinting Biotemplates and Excited by Radially Polarized Vector Beam" presents a new MEF configuration. The substrate's MEF performance was evaluated, and the results showed more than 10 times stronger fluorescence intensity with RVB excitation compared to no substrate and about 2 times stronger with RVB than with linearly polarized beam. However, the paper requires significant revisions and is not suitable for publication in its current form.

Major Corrections:

1.    The authors are encouraged to approach the writing of their manuscript with a great deal of care, particularly in regard to its intended audience. As the manuscript is directed towards a general audience, it is crucial that the language and phrasing used throughout are clear and easily understood. To this end, the authors should endeavor to compose brief, concise sentences that convey their message effectively. Lengthy sentences, while occasionally useful in other contexts, are not recommended in this case, as they can make it more challenging for readers to follow the authors' line of thought and understand their findings. It is recommended that the authors review their manuscript carefully, paying close attention to passages where sentences are lengthy or convoluted, and revise these portions accordingly. By doing so, they will ensure that their work is presented in a manner that is accessible and easy to understand for a broad readership. 

Following are the line numbers which should be reworked: Line number: 16-19, 39-43, 53-56, 58-61, 63-67, 71-73, 73-78, 81-84, 86-88, 88-91,116-119, 152-154, 159-161, 190-192, 193-196, and 196-198.

The authors are advised that specific lines in their manuscript require significant revision. Therefore, they are requested to review these lines carefully and make the necessary alterations to improve clarity and precision.

2.    It has been observed that the authors have omitted suitable references in several instances throughout their manuscript. As references are a crucial component of good scientific practice, it is essential that they are included wherever necessary. Therefore, the authors are advised to carefully review their manuscript and ensure that all relevant references are appropriately cited. 

For example. Line number 45-47 requires a reference, line 57-58, line 61-63, and line 152-154.

3.    Line21: Abstract – “In addition, the prepared substrate is verified 20 to have good uniformity.”  The authors have used the term "uniformity" without specifying what it refers to. To avoid ambiguity, they should clarify which aspect of their research they are referring to when using the term "uniformity."

4.    Line 51: Could you please consider substituting "Up to now" with a different word or phrase, as it does not suit the context of the sentence?

5.    Line 71-78: The sentences used in the manuscript are identical to those presented in the abstract. It is advised that the authors avoid repeating the same sentences throughout the manuscript and instead use alternative phrasing. In light of this, it is recommended that the authors provide a concise overview of the experiments, findings, and any associated advantages and disadvantages in the manuscript, rather than merely restating the same sentences used in the abstract. By doing so, the authors will provide a more detailed and informative account of their research that will be of greater value to readers.

6.    Line 94, Line100: The authors have referred to G1 and G2 in the manuscript, but these elements have not been labeled in Figure 1. To ensure consistency and avoid confusion for readers, it is recommended that G1 and G2 be clearly marked in the figure. Otherwise, it is advisable to refrain from mentioning components that are not depicted in the figure to prevent discrepancies in the information presented.

7.    Line 101: The authors have used the term "slightly" without providing a reference or comparison point. As the term is subjective and open to interpretation, it is recommended that the authors provide a clear reference or comparison to contextualize their use of the term.

8.    Figure 1: The authors are advised to ensure that there is no overlapping of text with the figure elements or components in order to improve the clarity of the figure. Additionally, it is recommended to use text of the same color as the component being referred to, for example, "UV curing adhesive" in Figure 1a could be in the same color as the blue component. To enhance the readability of the figure, it is suggested that the arrows connecting Figure 1a to 1b, 1c, 1d, and 1e be a different color that can be easily distinguished from the background. The positioning of Figure 1e should be brought down and not placed in the same plane as 1c, to avoid confusion for readers. As a reader, it was difficult to follow the arrows due to their similar color as the background and it took a while to locate Figure 1e.

9.    Line 115: It is recommended to abbreviate the term "SEF" before it is used in the manuscript, as it has not been previously abbreviated.

10. Line 126, “Fx” has not been shown in Figure 2. 

11. Line 134: Please substitute the word "bigger" with "larger".

12. Line 137: Kindly include the numerical aperture (NA) value of the micro-objective, similar to how it was done for the other objective in line 133.

13. Line 159, Line 166: Could the authors please provide a detailed explanation of the terms 'oblique view' and 'top-down view' as used in the manuscript? As these terms may not be familiar to a general audience, it would be beneficial to elaborate on their meanings in order to convey the message more clearly and concisely. Although these terms may be common within the SEM user community, it is important to keep in mind that the manuscript is also intended for a broader readership.

14. Line 198: "Considering the use of the word "can" in the sentence, it raises uncertainty about the enhancement. May the authors provide further clarification or evidence to support the assertion?"

15. Figure 4: It is recommended that Figures 4a and 4b are combined in order to provide a clearer distinction between the signal from glass compared to that from LPB and RVP. The current presentation may cause confusion among readers as to which signal is being referred to. Combining the figures will allow for a more comprehensive understanding of the data and prevent any potential misunderstandings.

Figure 4: Could the authors kindly clarify whether the data presented in Figure 4d are accompanied by any estimates of measurement uncertainty, such as error bars or confidence intervals? If such estimates are available, it is suggested that they be included in the figure, as this is a common practice in scientific data visualization and can provide readers with valuable information about the precision and accuracy of the measureme

Author Response

Thanks for your helpful advice. Please see the attachment.

Author Response

Thanks for your helpful advice. We have replied to your comments point-by-point. Please see the attachment.

Reviewer 3 Report

This is a good manuscript. It may be published after a minor revision. In the revised version the authors should address the following issues:

lines 75-77 (copied in the lines 217-219 [?]) mention that the intensity of fluorescence increased only  two times. It is a rather modest result for such complex  experiments.  Possibly the authors may explore on that result.

Intensity of the fluorescence, not intensity of a a spectrum. The English should be corrected.

The title is too long, should be re-written.

UV-curable adhesive should be described in more details. Currently, the reader is puzzled: what adhesive, what is a source of UV, etc.

The Introduction section should be abridged at least twice; it is not a review article but a set up of a task.

The set of four words "Chinese cicada “meimuna mongolica” " is repeated in the manuscript ten times or so. It should be abbreviated as "the cicada" or another way and be explained, what is it in the Experimental part.

I recommend a reference to a book of J.R. Lakowicz on fluorescence as a ref. #1.

Author Response

Thanks for your helpful advice. We have replied to the comments point by point. Please see the attachment.

Reviewer 4 Report

The author has demonstrate a nanoimpronted plasmonic film for metal enhanced fluorescence. However , using radially polarized vector beam for metal enhanced fluorescence on nanostructures has been published in other works before. In addition, the author might need to do more investigation on how different parameters affect the phenomenon on the metal enhancedd fluorescence with the bio template imprinted nanostructures, for example, coating materials, thinkness, excitation power, wavelength, fluorophores and so on. Over all, although the author clearly expressed their ideas, the author might try to demonstrate the novelty or the improvement in current technology/theory to enhance the impact of this manuscript. 

Author Response

Thans for your helpful advice. We have replied to the comments point-by-point. Please see the attachment.

Round 2
